# High-Efficient Calculation Method for Sensitive PDGEs of Five-Axis Reconfigurable Machine Tool

Zhanqun Song [1], Shuang Ding [1,2,*], Zhiwei Chen [1], Zhongwang Lu [1] and Zhouzhou Wang [1]

[1] College of Mechanical Engineering, Yangzhou University, Yangzhou 225127, China; songzhanqun1@163.com (Z.S.); heidaxiaohei@163.com (Z.C.); 13196468788@163.com (Z.L.); dfwangzhouzhou@163.com (Z.W.)
[2] Jiangsu Key Laboratory of Digital Manufacturing for Industrial Equipment and Control Technology, Nanjing 211816, China
* Correspondence: sding@yzu.edu.cn

**Abstract:** Sensitive geometric errors of a machine tool have significant influence on machining accuracy, and it is important to identify them. Complex modeling and analysis must be carried out to identify the sensitive geometric errors of a five-axis machine tool by using the traditional method. Once the configuration structure of the machine tools is reconstructed, repetitive error modeling and analysis are required, and the identification cycle of sensitive geometric errors is long. Therefore, this paper proposes a high-efficient calculation method for sensitive position-dependent geometric error (PDGEs) identification of a five-axis reconfigurable machine tool. According to the results of sensitive geometric errors of the RTTTR-type and TTTRR-type five-axis machine tools, the mapping expressions between sensitive PDGEs and the configuration structure of machine tools was established. In this method, sensitive PDGEs can be calculated directly according to the mapping expression, which eliminates the process of error modeling and analysis. Taking a RTTTR-type five-axis machine tool as an example, the sensitive PDGEs were calculated according to the presented mapping expressions without error modeling and analysis. A series of analysis points in the machining area were selected to compare the machining errors before and after sensitive PDGE compensation. The results show that this calculation method is accurate.

**Keywords:** sensitive geometric errors identification; reconfigurable machine tool; position-dependent geometric errors; high-efficient calculation method

## 1. Introduction

Five-axis computer numeric controlled (CNC) machine tools play a leading role in the complex surface manufacturing of high-end parts, and precision is one of the key indexes to evaluate its performance. The main errors that affect the machining accuracy of machine tools include geometric errors, thermal errors, and cutting force errors. In particular, geometric errors account for more than 50% of the total errors [1], which have considerable influence on the machining accuracy, and therefore, deserve more attention. It is of great significance to recognize and control the sensitive geometric errors through precision design and error compensation.

The reconfigurable machine tool was first proposed by Professor Koren in the University of Michigan [2]. By changing the structure module of the machine tool, the transformation and replacement of products and machine tools were realized. In order to adapt to the challenge of economic benefit and processing efficiency brought by the change of processing demand, customized and flexible services of reconfigurable machine tools are provided. Many scholars have carried out a lot of work on the design of reconfigurable machine tools, optimization of configuration, and reconfigurable research of the numerical control system [3,4]. The machine tool can be automatically reconfigured to a different

structure by adding, deleting, or changing the machine tool modules when the processing requirements change [5].

After the reconstruction of a five-axis machine tool, the same parts are reorganized into different structures. Change in action direction of the sensitive geometric errors leads to a change in machining accuracy of a machine tool. It is necessary to analyze the sensitive geometric errors of the reconstructed machine tool for high-efficient precision design and error compensation. If the sensitive geometric errors can be calculated quickly according to the machine tool structure, design and optimization of the structure of a reconfigurable machine tool can be conducted to avoid or to reduce the influence of sensitive geometric errors. In this way, the cost for accuracy improvement of the reconstructed machine tool can be significantly reduced. In the existing literature, complex error modeling and analysis are usually implemented for each machine tool structure to identify the sensitive geometric errors, which are unable to satisfy the rapid response requirements. Therefore, an efficient algorithm is urgently needed to identify sensitive geometric errors directly according to the configuration of a machine tool.

Sensitivity error analysis of machine tools can be used to determine the most critical errors on machine accuracy. The sensitivity error analysis methods of machine tools are usually divided into local sensitivity analysis (LSA) and global sensitivity analysis (GSA) [6]. Both of them are based on spatial geometric error modeling. LSA emphasizes the impact of a single parameter and can only judge the sensitivity of one processing point at a time. Fan et al. established the sensitivity model of a machine tool based on the first-order sensitivity analysis method [7]. Li et al. used the LSA method to analyze the error sensitivity of the spatial accuracy [8]. Chen et al. established an error model of an RTTTR-type five-axis machine tool based on the MBS method. The error values were uniformly set as determinate values, and the LSA method was used to analyze the sensitivity of the error [9]. CHENG et al. established the sensitivity matrix of a four-axis precision machine tool based on the matrix differential method and used the LSA method to identify the key geometric errors of the machine tool [10]. Yao et al. analyzed the local sensitivity of the errors based on the multiple linear regression method [11].

GSA considers the coupling relationship, which can determine the influence intensity of each error term and strength of interaction among various error terms. Cheng et al. used the Sobol global sensitivity analysis method to determine the critical geometric errors of machine tools [12]. Guo et al. used the first-order sensitivity analysis method to identify the sensitivity error of the five-axis machine tool and conducted a global sensitivity analysis of the machine tool based on an extend Fourier amplitude sensitivity test (EFAST) method [13]. Cheng et al. conducted a global sensitivity analysis of the error of a five-axis machine tool based on the EFAST method [14]. Cheng et al. conducted a global sensitivity analysis of machine tool errors based on the product of exponential (POE) screw theory and the Morris sensitivity analysis method [15]. Fu et al. established the error sensitivity matrix for each axis based on POE theory and evaluated the influence degree of each axis on the errors [16]. Xia et al. used the Morris global sensitivity analysis method to quantify the error sensitivity and identified the key errors and sensitive parts of the five-axis gear grinding machine [17]. Guo et al. modeled the error of a five-axis machine tool based on the MBS method and analyzed the sensitivity of the error model [18]. Li et al. considered the position and attitude errors of the tool and improved the traditional sensitivity analysis method by simplifying the output number of error sensitivity [19]. Zhang et al. used the multiplicative dimensional reduction method (M-DRM) to analyze the sensitivity of machine tools [20]. Zou et al. used the Sobol global sensitivity analysis method to analyze the error sensitivity of the three-axis diamond lathe [21]. Fan et al. developed a quantitative interval sensitivity analysis method (QISA) to calculate the sensitivity of geometric errors [22]. Li et al. defined a new machine tool error sensitivity analysis method by expressing the tool path and path error on a tool coordinate system [23]. Li et al. simplified the expression form of the machine tool error sensitivity index and proposed a new index to express error fluctuations [24].

At present, GSA is widely used in error sensitivity analysis. Among various methods, the Morris global sensitivity analysis is mainly used to analyze the impact of input parameters on output parameters when input parameters change in the global scope. The five-axis machine tool has a large number of PDGEs, and there is a coupling relationship between the error terms. The Morris global sensitivity analysis method, based on variable discrete and random sampling, can effectively analyze the sensitivity of the PDGEs of five-axis machine tools.

Although both LSA and GSA are effective methods for sensitive geometric error identification, they need error modeling and analysis process based on HTM, which has a low response speed when structure is reconfigured. Therefore, this paper tries to find a new way to calculate the sensitive geometric errors according to the machine tool structure without error modeling and analysis. The mapping expressions between the sensitive geometric errors and the machine tool structure are established and organized.

According to the different structural configurations of the five-axis machine tools, the sensitive PDGEs were analyzed by using the traditional error modeling method. Through the systematic analysis for different structures of RTTTR and TTTRR-type five-axis machine tools, the mapping expression between the sensitive PDGEs and the configuration of machine tools was established. Additionally, a high-efficient calculation method of sensitive PDGEs was proposed according to the mapping expressions without error modeling and analysis. The proposed method can directly identify sensitive PDGEs according to the configuration structure of the machine tool without requiring theoretical modeling and sensitivity analysis, which is suitable for quick error analysis of a machine tool.

The following sections are divided into four parts to display the contents of this article. Section 2 introduces the theoretical basis of the high-efficient calculation method. In Section 3, the high-efficient calculation method of sensitive PDGEs is introduced. In Section 4, the related simulation is described. Section 5 summarizes the results of this paper.

## 2. Theoretical Basis of the High-Efficient Calculation Method

This section introduces the theoretical basis of the high-efficient calculation method. Based on the traditional analysis method in this section, the RTTTR and TTTRR-type five-axis machine tools with different structures were analyzed. Through the systematic analysis for different structures of machine tools, eventually the mapping expressions between the sensitive PDGEs and the configuration of machine tools was established.

### 2.1. Morris Global Sensitivity Analysis Method

There have been a total of 36 PDGEs of the five-axis reconfigurable machine tool. The motion axes of a five-axis machine tool consist of *X*, *Y*, *Z*, *A*, *B* or *X*, *Y*, *Z*, *A*, *C* and so on. The names and geometric meanings of error are shown in Table 1.

**Table 1.** Definition of PDGEs.

| The Name of the Linear Error Term | The Geometric Meaning of the Error Term | The Name of the Angular Error Term | The Geometric Meaning of the Error Term |
|---|---|---|---|
| $\Delta x(X)$ | Positioning error of the $X$ axis in the $x$ direction | $\Delta \alpha(X)$ | Rolling error of $X$ axis in $x$ direction |
| $\Delta y(X)$ | Straightness error of $X$ axis in $y$ direction | $\Delta \beta(X)$ | $X$ axis angular error in $y$ direction |
| $\Delta z(X)$ | Straightness error of $X$ axis in $z$ direction | $\Delta \gamma(X)$ | $X$ axis angular error in $z$ direction |
| $\Delta x(Y)$ | Straightness error of $Y$ axis in $x$ direction | $\Delta \alpha(Y)$ | $Y$ axis angular error in $x$ direction |
| $\Delta y(Y)$ | Positioning error of $Y$ axis in $y$ direction | $\Delta \beta(Y)$ | Rolling error of $Y$ axis in $y$ direction |
| $\Delta z(Y)$ | Straightness error of $Y$ axis in $z$ direction | $\Delta \gamma(Y)$ | $Y$ axis angular error in $z$ direction |
| $\Delta x(Z)$ | Straightness error of $Z$ axis in $x$ direction | $\Delta \alpha(Z)$ | $Z$ axis angular error in $x$ direction |
| $\Delta y(Z)$ | Straightness error of $Z$ axis in $y$ direction | $\Delta \beta(Z)$ | $Z$ axis angular error in $y$ direction |
| $\Delta z(Z)$ | Positioning error of $Z$ axis in $z$ direction | $\Delta \gamma(Z)$ | Rolling error of $Z$ axis in $z$ direction |
| $\Delta x(A)$ | Positioning error of the $A$ axis in the $x$ direction | $\Delta \alpha(A)$ | Rolling error of $A$ axis in $x$ direction |
| $\Delta y(A)$ | Straightness error of $A$ axis in $y$ direction | $\Delta \beta(A)$ | $A$ axis angular error in $y$ direction |
| $\Delta z(A)$ | Straightness error of $A$ axis in $z$ direction | $\Delta \gamma(A)$ | $A$ axis angular error in $z$ direction |
| $\Delta x(B)$ | Straightness error of $B$ axis in $x$ direction | $\Delta \alpha(B)$ | $B$ axis angular error in $x$ direction |
| $\Delta y(B)$ | Positioning error of $B$ axis in $y$ direction | $\Delta \beta(B)$ | Rolling error of $B$ axis in $y$ direction |
| $\Delta z(B)$ | Straightness error of $B$ axis in $z$ direction | $\Delta \gamma(B)$ | $B$ axis angular error in $z$ direction |
| $\Delta x(C)$ | Straightness error of $C$ axis in $x$ direction | $\Delta \alpha(C)$ | $C$ axis angular error in $x$ direction |
| $\Delta y(C)$ | Straightness error of $C$ axis in $y$ direction | $\Delta \beta(C)$ | $C$ axis angular error in $y$ direction |
| $\Delta z(C)$ | Positioning error of $C$ axis in $z$ direction | $\Delta \gamma(C)$ | Rolling error of $C$ axis in $z$ direction |

This section takes the RTTTR-type five-axis machine tool as an example, as seen in Figure 1. Sub-coordinate system on each axis of the machine tool was established. Based on MBS theory [25,26] and HTM, the geometric error model of the five-axis machine tool was established. The Morris global sensitivity analysis method [27] was used to analyze sensitive PDGEs of the machine tool.

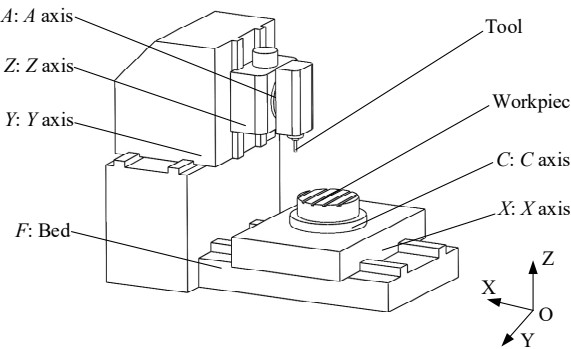

**Figure 1.** RTTTR-type five-axis machine tool structure diagram.

The cutting point coordinate in the tool coordinate system is set as $\begin{bmatrix} 0 & 0 & 0 & 1 \end{bmatrix}^T$. The machine tool is divided into two chains according to the topological relation. The workpiece chain is bed-*X-C*-workpiece, and the tool chain is bed-*Y-Z-A*-tool. Supposing that the ideal position transformation matrix between adjacent body *i* and *j* is $T_{ijp}$, among them, $i \in (F, X, Y, Z, A, C)$ and $j \in (X, Y, Z, A, C)$, the *F*, *X*, *Y*, *Z*, *A* and *C* represent bed, *X* axis, *Y* axis, *Z* axis, *A* axis, and *C* axis, respectively. The position error transformation matrix between adjacent body *i* and *j* is $\Delta T_{ijp}$. The ideal motion transformation matrix of adjacent body is $T_{ijs}$. The motion error transformation matrix of adjacent body is $\Delta T_{ijs}$. Then, the ideal forming matrix of the machine tool is shown in Equation (1), and the actual forming matrix of the machine tool is shown in Equation (2). The comprehensive error matrix of the machine tool can be given as Equation (3).

$$P_{ideal} = (T_{FXp}T_{FXs}T_{XCp}T_{XCs})^{-1}T_{FYp}T_{FYs}T_{YZp}T_{YZs}T_{ZAp}T_{ZAs}\begin{bmatrix} 0 & 0 & 0 & 1 \end{bmatrix}^T \quad (1)$$

$$P_{actual} = (T_{FXp}\Delta T_{FXp}T_{FXxs}\Delta T_{FXs}T_{XCp}\Delta T_{XCp}T_{XCs}\Delta T_{XCs})^{-1}T_{FYp}\Delta T_{FYp}T_{FYs}\Delta T_{FYs}T_{YZp}\Delta T_{YZp}T_{YZs}\Delta T_{YZs}$$
$$T_{ZAp}\Delta T_{ZAp}T_{ZAs}\Delta T_{ZAs}\begin{bmatrix} 0 & 0 & 0 & 1 \end{bmatrix}^T \quad (2)$$

$$E = P_{actual} - P_{ideal} = \begin{bmatrix} E_x & E_y & E_z & 0 \end{bmatrix}^T \quad (3)$$

The Morris global sensitivity analysis method can effectively sort and evaluate the influence of model parameters; it has good applicability for models with many analysis parameters. Based on the experimental design of the one-time change method, only one model parameter is changed for each sampling, and the influence of each model parameter is calculated in turn so as to realize the evaluation of the sensitivity of the model parameters. The specific form is as follows.

$$EE_i = \frac{y(x_1, \ldots, x_{i1}, \ldots, x_n) - y(x_1, \ldots, x_{i1} + \Delta, \ldots, x_n)}{\Delta} \quad (4)$$

In Equation (4), $EE_i$ is the influence effect of the *i*th input parameter, *y* is the output function, *x* is the input parameter, and $\Delta$ is the variation of the input parameter $\Delta = 1/(q-1)$, *q* is the number of samples for each error term. The number of cyclic sam-

pling is set to *SN*, and the influence effect of each input parameter is calculated in turn. In this way, the *SN* effects of each input parameter can be obtained.

$$\mu_i = \frac{1}{SN} \sum_{j=1}^{SN} EE_{ij}$$
$$\sigma_i = \sqrt{\frac{1}{SN} \sum_{j=1}^{SN} \left(EE_{ij} - \mu_i\right)^2}$$

(5)

As shown in Equation (5), the mean $\mu_i$ and standard deviation $\sigma_i$ of each effect were calculated, respectively. Where $\mu_i$ reflects the influence of each input parameter on the output function; the greater the value, the greater the influence. $\sigma_i$ reflects the coupling strength between each input parameter and other parameters; the larger the value, the higher the coupling strength.

The 30 PDGEs of the five-axis machine tool shown in Figure 1 were used as input parameters, and the comprehensive error model in Equation (3) was taken as output function by using the Morris method. The range of linear errors in PDGEs was set as [0, 15] μm and for angular errors was set as [0, 0.015] deg. One-hundred and twenty (*SN*) cycles of sampling were conducted in the working area to obtain the mean value $\mu_i$ and standard deviation $\sigma_i$, so as to identify the sensitive PDGEs of the machine tool.

The sensitivity results are shown in Figure 2a–c. Serial numbers 1–6 represent the PDGEs of *X* axis ($\Delta x(X)$, $\Delta y(X)$, $\Delta z(X)$, $\Delta\alpha(X)$, $\Delta\beta(X)$, and $\Delta\gamma(X)$), serial numbers 7–12 represent the PDGEs of *Y* axis($\Delta x(Y)$, $\Delta y(Y)$, $\Delta z(Y)$, $\Delta\alpha(Y)$, $\Delta\beta(Y)$, and $\Delta\gamma(Y)$), serial numbers 13–18 represent the PDGEs of *Z* axis ($\Delta x(Z)$, $\Delta y(Z)$, $\Delta z(Z)$, $\Delta\alpha(Z)$, $\Delta\beta(Z)$, and $\Delta\gamma(Z)$), serial numbers 19–24 represent the PDGEs of *A* axis ($\Delta x(A)$, $\Delta y(A)$, $\Delta z(A)$, $\Delta\alpha(A)$, $\Delta\beta(A)$, and $\Delta\gamma(A)$) and serial numbers 25–30 represent the PDGEs of *C* axis ($\Delta x(C)$, $\Delta y(C)$, $\Delta z(C)$, $\Delta\alpha(C)$, $\Delta\beta(C)$, and $\Delta\gamma(C)$), respectively. The red point represents the mean value $\mu_i$ of the influence effect of each error term and reflects the sensitivity of each error term. The blue point $\sigma_i$ represents the coupling strength between each error term and other error terms. In this way, the sensitive PDGEs of the machine tool in three coordinate directions can be obtained.

### 2.2. Discovery of Mapping Relationships

Based on the error modeling and sensitive PDGEs analysis method described in Section 2.1, the sensitive PDGEs analyses of RTTTR and TTTRR-type five-axis reconfigurable machine tools with different structures were carried out, respectively. After comparing the analysis results of sensitive PDGEs with the structure of the machine tool, it is found that there is a mapping relationship between the sensitive PDGEs of the machine tool and the configuration of each axis of the machine tool. Some of these relationships are described below.

Case 1: The linear positioning errors of the translational axis corresponding to the identification direction are always sensitive PDGEs [14,24]. Examples are described as follows: in *x* direction, sensitive linear PDGEs of translational axes are $\Delta x(X)$, $\Delta x(Y)$, and $\Delta x(Z)$. In *y* direction, sensitive linear PDGEs of translational axes are $\Delta y(X)$, $\Delta y(Y)$, and $\Delta y(Z)$. In *z* direction, sensitive linear PDGEs of translational axes are $\Delta z(X)$, $\Delta z(Y)$, and $\Delta z(Z)$. The definition of each error term is shown in Table 1.

Case 2: If the translational axis belongs to the workpiece chain, the axis has two sensitive angular PDGEs in each identification direction. The examples are as follows: assuming that *Y* axis belongs to workpiece chain, in *x* direction, the sensitive angular PDGEs of *Y* axis are $\Delta\beta(Y)$ and $\Delta\gamma(Y)$; in *y* direction, the sensitive angular PDGEs of *Y* axis are $\Delta\alpha(Y)$ and $\Delta\gamma(Y)$; in *z* direction, the sensitive angular PDGEs of *Y* axis are $\Delta\alpha(Y)$ and $\Delta\beta(Y)$.

There are other relationships besides those mentioned above. In order to facilitate the identification, this paper presents these mapping relations in the form of mathematical expressions in Section 3. Combined with these mapping expressions, the sensitive PDGEs

of the machine tool can be directly obtained through simple calculations.

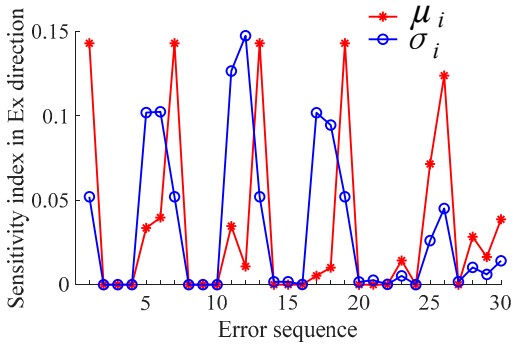

(**a**) Sensitivity index for $E_x$

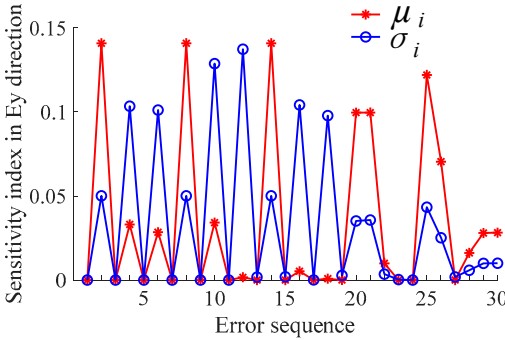

(**b**) Sensitivity index for $E_y$

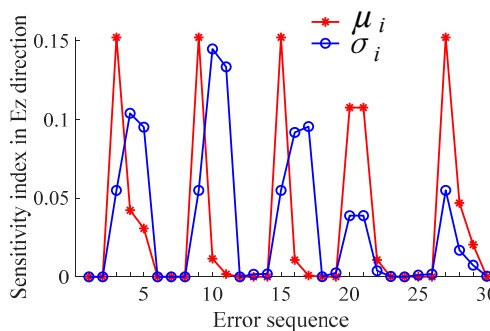

(**c**) Sensitivity index for $E_z$

**Figure 2.** The sensitivity index of the RTTTR-type machine tool.

## 3. Mapping Expressions

This section describes the mapping expression between the sensitive PDGEs of the five-axis machine tool and the machine tool structure. Based on this mapping expression, the sensitive PDGEs of the five-axis reconfigurable machine tool can be quickly identified, eliminating the process of error modeling and analysis.

The traditional process of sensitive geometric error analysis is shown in Figure 3a, and the proposed method is shown in Figure 3b. It can be found that the identification process of the sensitive geometric error of a single-structure five-axis machine tool is inherently complicated. Once the structure of the machine tool is reorganized, complex error modeling and analysis need to be performed again. In contrast with traditional methods, the proposed method can not only quickly identify the sensitive PDGEs of a specific five-axis machine tool but also quickly respond to the reorganized five-axis machine tool.

Before introducing the proposed method, the relevant symbols are defined and explained. This article uses lowercase italics when referring to directions, such as: *x*, *y*, and *z* and when referring to axes, uses uppercase italics, such as *X, Y, Z, A, B,* and *C*. Other related expressions are described in Sections 3.1–3.3, which describe the high-efficient calculation methods for sensitive PDGEs identification of RTTTR and TTTRR-type five-axis reconfigurable machine tools, respectively.

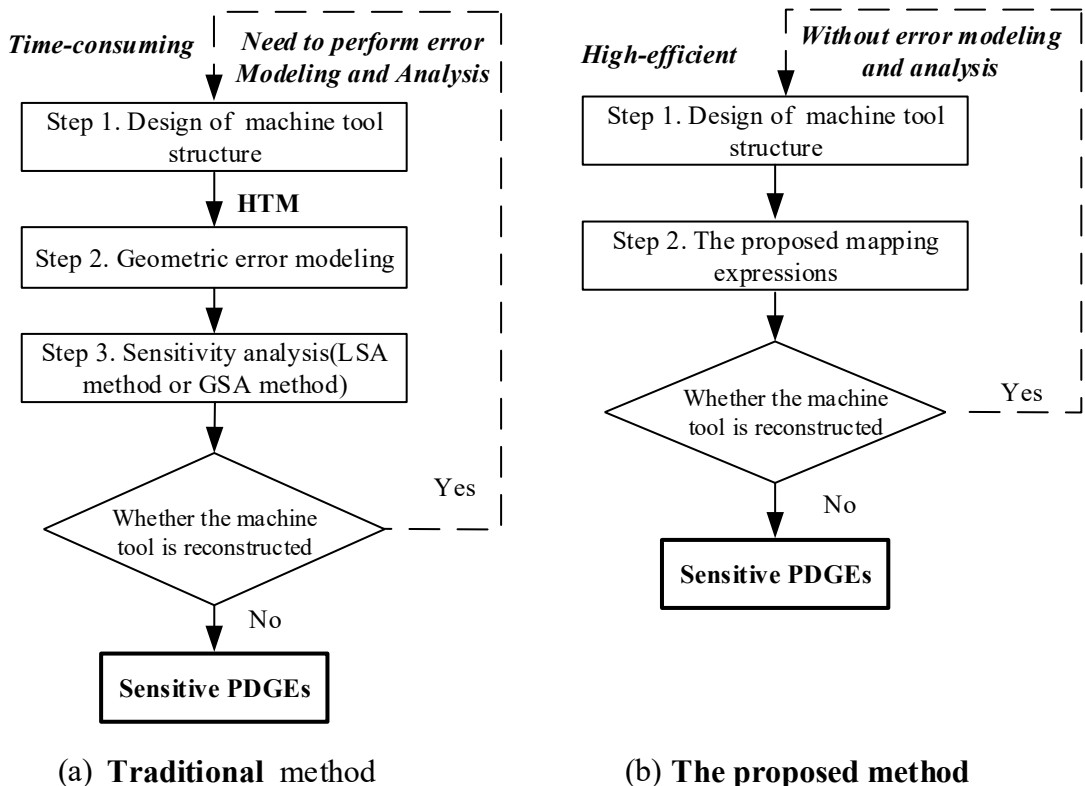

(a) **Traditional** method          (b) **The proposed method**

**Figure 3.** The process of identifying the sensitive geometric errors of the five-axis reconfigurable machine tool.

### 3.1. The Defined Symbols and Expressions

The new definition of mapping expressions of sensitive PDGEs identification is shown as Equation (6).

$$Typ(i)_{Axis}^{Dir} = \Delta Cal(parm)(Axis) \tag{6}$$

Where superscript *Dir* denotes the identification direction. Generally, the identification directions of spatial positional errors of a machine tool are *x*, *y*, and *z* directions along *X*-axis, *Y*-axis, and *Z*-axis, respectively. *Axis* denotes the motion axis of a machine tool, $Axis \in (X, Y, Z, A, B, C)$. $i = 1, 2, 3,$ or 4, which is used to distinguish different identification axes and different error types. *Cal* is the mapping algorithm; the type and direction of error can be obtained by *Cal* calculating *parm*. The detailed meanings of the defined symbols are described as follows:

(1)    $Typ(i)_{Axis}^{Dir}$ represents the sensitive PDGEs of the motion axis (*Axis*) in the identification direction (*Dir*). $Typ(i)$ is the type of sensitive geometric error to be calculated and is divided into the following four categories:$Typ(1)$ represents the sensitive linear PDGEs of the translational axis. $Typ(2)$ represents the sensitive linear PDGEs of the rotational axis. $Typ(3)$ represents the sensitive angular PDGEs of the translational axis. $Typ(4)$ represents the sensitive angular PDGEs of rotational axis.

The sensitive PDGEs of each axis in each direction are identified in *x*, *y,* and *z* directions; *Dir* is the direction of error identification, $Dir \in (x, y, z)$.

*Axis* indicates the motion axis that needs to be identified. Once $Typ(i)$ is defined, the selection range of the corresponding axis is also determined. If $i = 1$ or 3, $Axis \in (X, Y, Z)$. If $i = 2$ or 4, $Axis \in (A, B, C)$.

For example, $Typ(1)_Y^x$ represents the sensitive linear PDGEs of the $Y$ axis in $x$ direction. $Typ(2)_A^z$ represents the sensitive linear PDGEs of the $A$ axis in $z$ direction. $Typ(3)_Y^x$ represents the sensitive angular PDGEs of the $Y$ axis in $x$ direction. $Typ(4)_A^z$ represents the sensitive angular PDGEs of the $A$ axis in $z$ direction.

(2)　The defined mapping expressions $\Delta Cal(parm)(Axis)$ is used to calculate the error items in Table 1, taking $\Delta x(X)$ for an example, where $Cal(parm) = x$ and $Axis = X$.

In $\Delta Cal(parm)(Axis)$, $Axis \in (X, Y, Z, A, B, C)$; $parm \in (Dir, Axis, Toolvector)$, *Toolvector* is the tool axis vector (usually in $z$ direction); when calculating sensitive linear PDGEs, $Cal(parm)(x, y, z)$; when calculating the sensitive angular PDGEs, $Cal(parm) \in (\alpha, \beta, \gamma)$. Taking $X$ axis as an example, the calculation results of $\Delta Cal(parm)(X)$ are shown in Appendix A. The calculation results of other axes ($Y$, $Z$, $A$, $B$, or $C$ axis) can also be obtained in this form.

In Appendix A, in order to facilitate understanding, the result of the $Cal(parm)$ in $Cal(parm)(X)$ is marked with two underscores. Among them, $Cal(parm)$ represents the axial direction of *parm*; $Cal(\overline{parm})$ represents the axial direction of the other two directions excepting the direction of *parm*; $Cal(parm_1 + parm_2)$ represents the axial direction of the $parm_1$ and $parm_2$; $Cal(\overline{parm_1 + parm_2})$ indicates the axial direction excepting $parm_1$ and $parm_2$.

(3)　Combined with the relevant definitions in (1) and (2), the relevant calculation forms of Equation (6) are illustrated with examples as follows:
If Equation (6) is $Typ(1)_X^z = \Delta Cal(z)(X)$, then $Typ(1)_X^z = \Delta z(X)$;
If Equation (6) is $Typ(3)_X^z = \Delta Cal(z)(X)$, then $Typ(3)_X^z = \Delta\gamma(X)$;
If Equation (6) is $Typ(2)_A^y = \Delta Cal(\overline{y})(A)$, then $Typ(2)_A^y = \{\Delta x(A), \Delta z(A)\}$;
If Equation (6) is $Typ(4)_A^y = \Delta Cal(\overline{y})(A)$, then $Typ(4)_A^y = \{\Delta\alpha(A), \Delta\gamma(A)\}$.

### 3.2. Calculation Method of RTTTR-Type Five-Axis Machine Tools

This section introduces the mapping expressions for sensitive PDGE identification of the RTTTR-type five-axis reconfigurable machine tool. According to the configuration axis information of the machine tool, combined with the basic calculation formula described in Section 3.1, the sensitive PDGEs can be quickly obtained by a simple calculation.

This section takes the machine tool shown in Figure 1 as an example to illustrate the proposed method. In Figure 1, the workpiece chain is: bed-*X*-*C*-workpiece, the tool chain is: bed-*Y*-*Z*-*A*-tool. The rotational axis in the workpiece chain is marked as $R_1$, the rotational axis in the tool chain is marked as $R_2$, the translational axis closest to the bed in the tool chain is marked as $T_F$, and the axis vector of tool is $z$ direction.

#### 3.2.1. Sensitive Linear PDGEs for Translational Axes

If the sensitive linear PDGEs of translational axes are identified, $i = 1$, $Axis \in (X, Y, Z)$ can be obtained according to Section 3.1. Where $Dir \in (x, y, z)$, then the mapping expression is given as:

$$Typ(1)_{Axis}^{Dir} = \Delta Cal(Dir)(Axis) \tag{7}$$

According to Equation (7), take the $X$ axis as an example:
If sensitivity PDGE in $x$ direction is identified, then $Typ(1)_X^x = \Delta Cal(x)(X) = \Delta x(X)$ can be obtained;
If sensitivity PDGE in $y$ direction is identified, then $Typ(1)_X^y = \Delta Cal(y)(X) = \Delta y(X)$ can be obtained;
If sensitivity PDGE in $z$ direction is identified, then $Typ(1)_X^z = \Delta Cal(z)(X) = \Delta z(X)$ can be obtained. In the same way, the sensitive linear PDGEs of the $Y$ axis and $Z$ axis can be calculated.

### 3.2.2. Sensitive Linear PDGEs for Rotational Axes

If the sensitive linear PDGEs of rotational axes are identified, $i = 2$, $Axis \in (A, B, C)$ can be obtained according to Section 3.1. Where $Dir \in (x, y, z)$, then the mapping expression is given as:

$$\begin{cases} \text{if :} Cal(Dir) = Cal(Axis), \text{then :} Typ(2)_{Axis}^{Dir} = \Delta Cal(Axis)(Axis) \\ \text{if :} Cal(Dir) \neq Cal(Axis), \text{then :} Typ(2)_{Axis}^{Dir} = \Delta Cal(\overline{Axis})(Axis) \end{cases} \tag{8}$$

According to Equation (8), take the $A$ axis as an example:

If sensitivity PDGE in $x$ direction is identified, $Cal(x) = Cal(A) = (Cal(Dir) = Cal(Axis))$, then $Typ(2)_A^x = \Delta Cal(A)(A) = \Delta x(A)$ can be obtained;

If sensitivity PDGE in $y$ direction is identified, $Cal(y) \neq Cal(A)(Cal(Dir) \neq Cal(Axis))$, then $Typ(2)_A^y = \Delta Cal(\overline{A})(A) = \{\Delta y(A), \Delta z(A)\}$ can be obtained;

If sensitivity PDGE in $z$ direction is identified, $Cal(z) \neq Cal(A)(Cal(Dir) \neq Cal(Axis))$, then $Typ(2)_A^z = \Delta Cal(\overline{A})(A) = \{\Delta y(A), \Delta z(A)\}$, can be obtained.

Therefore, the sensitive linear PDGEs of $A$ axis in the $x$ direction is $\Delta x(A)$, in the $y$ direction are $\Delta y(A)$ and $\Delta z(A)$, and in the $z$ direction are $\Delta y(A)$ and $\Delta z(A)$. In the same way, the sensitive linear PDGEs of the other rotational axis ($C$ axis) in the $x$, $y$, and $z$ directions can be calculated.

### 3.2.3. Sensitive Angular PDGEs of Translational Axis

If the sensitive linear PDGEs of translational axes are identified, $i = 3$, $Axis \in (X, Y, Z)$ can be obtained according to Section 3.1. Where $Dir \in (x, y, z)$, then the mapping expression is given as:

(1) When $Axis$ belongs to the workpiece chain

$$Typ(3)_{Axis}^{Dir} = \Delta Cal(\overline{Dir})(Axis) \tag{9}$$

(2) When $Axis$ belongs to the tool chain

$$\begin{cases} \text{if :} Cal(Dir) = Cal(R_2), \text{then :} Typ(3)_{Axis}^{Dir} = \Delta Cal(\overline{R_2})(Axis) \\ \text{if :} Cal(Dir) \neq Cal(R_2), \text{then :} Typ(3)_{Axis}^{Dir} = \Delta Cal(R_2)(Axis) \\ \text{if : tool chain has 3 translational axes, and} Cal(T_F) \neq Cal(R_1 + R_2), \\ \text{then :} Typ(3)_{Axis}^{Dir} = \Delta Cal(\overline{Dir})(T_F) \end{cases} \tag{10}$$

According to Equation (9), take the $X$ axis as an example ($X$ axis belongs to the workpiece chain):

If sensitivity PDGE in $x$ direction is identified, then $Typ(3)_X^x = \Delta Cal(\overline{x})(X) = \{\Delta\beta(X), \Delta\gamma(X)\}$ can be obtained;

If sensitivity PDGE in $y$ direction is identified, then $Typ(3)_X^y = \Delta Cal(\overline{y})(X) = \{\Delta\alpha(X), \Delta\gamma(X)\}$ can be obtained;

If sensitivity PDGE in $z$ direction is identified, then $Typ(3)_X^z = \Delta Cal(\overline{z})(X) = \{\Delta\alpha(X), \Delta\beta(X)\}$ can be obtained.

Therefore, the sensitive angular PDGEs of $X$ axis in the $x$ direction are $\Delta\beta(X)$ and $\Delta\gamma(X)$; in the $y$ direction are $\Delta\alpha(X)$ and $\Delta\gamma(X)$; and in the $z$ direction are $\Delta\alpha(X)$ and $\Delta\beta(X)$.

According to Equation (10), take the $Y$ axis as an example ($Y$ axis belongs to the tool chain):

If sensitivity PDGE in $x$ direction is identified, $Cal(x) = Cal(A)(Cal(Dir) = Cal(R_2))$, then $Typ(3)_Y^x = \Delta Cal(\overline{A})(Y) = \{\Delta\beta(Y), \Delta\gamma(Y)\}$ can be obtained;

If sensitivity PDGE in $y$ direction is identified, $Cal(y) \neq Cal(A)(Cal(Dir) \neq Cal(R_2))$, then $Typ(3)_Y^y = \Delta Cal(A)(Y) = \Delta\alpha(Y)$ can be obtained;

If sensitivity PDGE in $z$ direction is identified, $Cal(z) \neq Cal(A)(Cal(Dir) \neq Cal(R_2))$, then $Typ(3)_Y^z = \Delta Cal(A)(Y) = \Delta\alpha(Y)$ can be obtained.

Therefore, the sensitive angular PDGEs of $Y$ axis in the $x$ direction are $\Delta\beta(Y)$ and $\Delta\gamma(Y)$; in the $y$ direction is $\Delta\alpha(Y)$; and in the $z$ direction is $\Delta\alpha(Y)$. In the same way, the sensitive angular PDGEs for the $Z$ axis of the tool chain in the $x$, $y$, and $z$ directions can be calculated.

### 3.2.4. Sensitive Angular PDGEs of Rotational Axis

If the sensitive linear PDGEs of rotational axes are identified, $i = 4$, $Axis \in (A, B, C)$ can be obtained according to Section 3.1. Where $Dir \in (x, y, z)$, then the mapping expression is given as:

(1)　　When $Axis = R_1$

$$\begin{cases} \text{if}: Cal(Dir) = Cal(R_1), \text{then}: Typ(4)_{Axis}^{Dir} = \Delta Cal(\overline{R_1})(R_1) \\ \text{if}: Cal(Dir) \neq Cal(R_1), \text{then}: Typ(4)_{Axis}^{Dir} = \Delta Cal(X + Y + Z)(R_1) \end{cases} \tag{11}$$

(2)　　When $Axis = R_2$

$$\begin{cases} \text{if}: Cal(Dir) = Cal(Toolvector), \text{then}: Typ(4)_{Axis}^{Dir} = \Delta Cal(R_2)(R_2) \\ \text{if}: Cal(Dir) \neq Cal(Toolvector), \text{then}: Typ(4)_{Axis}^{Dir} = \Delta Cal(\overline{Dir + Toolvector})(R_2) \end{cases} \tag{12}$$

According to Equation (11), take the $C$ axis as an example ($C$ axis is marked as $R_1$ axis):

- If sensitivity PDGE in $x$ direction is identified, $Cal(x) \neq Cal(C)(Cal(Dir) \neq Cal(R_1))$, then $Typ(4)_C^x = \Delta Cal(X + Y + Z)(C) = \{\Delta\alpha(C), \Delta\beta(C), \Delta\gamma(C)\}$ can be obtained;
- If sensitivity PDGE in $y$ direction is identified, $Cal(y) \neq Cal(C)(Cal(Dir) \neq Cal(R_1))$, then $Typ(4)_C^y = \Delta Cal(X + Y + Z)(C) = \{\Delta\alpha(C), \Delta\beta(C), \Delta\gamma(C)\}$ can be obtained;
- If sensitivity PDGE in $z$ direction is identified, $Cal(z) = Cal(C)(Cal(Dir) = Cal(R_1))$, then $Typ(4)_C^z = \Delta Cal(\overline{C})(C) = \{\Delta\alpha(C), \Delta\beta(C)\}$ can be obtained.

Therefore, the sensitive angular PDGEs of $C$ axis in the $x$ direction are $\Delta\alpha(C)$, $\Delta\beta(C)$, and $\Delta\gamma(C)$; in the $y$ direction are $\Delta\alpha(C)$, $\Delta\beta(C)$, and $\Delta\gamma(C)$; and in the $z$ direction are $\Delta\alpha(C)$ and $\Delta\beta(C)$.

According to Equation (12), take the $A$ axis as an example ($A$ axis is marked as $R_2$ axis):

- If sensitivity PDGE in $x$ direction is identified, $Cal(x) \neq Cal(Toolvector)(Cal(Dir) \neq Cal(Toolvector)$, then $Typ(4)_A^x = \Delta Cal(\overline{x + z})(A) = \Delta\beta(A)$ can be obtained;
- If sensitivity PDGE in $y$ direction is identified, $Cal(y) \neq Cal(Toolvector)(Cal(Dir) \neq Cal(Toolvector))$, then $Typ(4)_A^y = \Delta Cal(\overline{y + z})(A) = \Delta\alpha(A)$ can be obtained;
- If sensitivity PDGE in $z$ direction is identified, $Cal(z) = Cal(Toolvector)(Cal(Dir) = Cal(Toolvector))$, then $Typ(4)_A^z = \Delta Cal(A)(A) = \Delta\alpha(A)$ can be obtained.

Therefore, the sensitive angular PDGEs of $A$ axis in the $x$ direction is $\Delta\beta(A)$; in the $y$ direction is $\Delta\alpha(A)$; and in the $z$ direction is $\Delta\alpha(A)$.

In summary, all the sensitive geometric errors in the $x$, $y$, and $z$ directions are shown in Table 2.

**Table 2.** Sensitive PDGEs in all directions.

| Error Direction | Sensitive PDGEs |
|---|---|
| $x$ | $\Delta x(X), \Delta x(Y), \Delta x(Z), \Delta x(A), \Delta x(C), \Delta y(C), \Delta\beta(X), \Delta\gamma(X), \Delta\beta(Y), \Delta\gamma(Y), \Delta\beta(Z), \Delta\gamma(Z), \Delta\alpha(C), \Delta\beta(C), \Delta\gamma(C), \Delta\beta(A)$ |
| $y$ | $\Delta y(X), \Delta y(Y), \Delta y(Z), \Delta y(A), \Delta z(A), \Delta x(C), \Delta y(C), \Delta\alpha(X), \Delta\gamma(X), \Delta\alpha(Y), \Delta\alpha(Z),$ $\Delta\alpha(C), \Delta\beta(C), \Delta\gamma(C), \Delta\alpha(A)$ |
| $z$ | $\Delta z(X), \Delta z(Y), \Delta z(Z), \Delta y(A), \Delta z(A), \Delta z(C), \Delta\alpha(X), \Delta\beta(X), \Delta\alpha(Y), \Delta\alpha(Z), \Delta\alpha(C), \Delta\beta(C), \Delta\alpha(A)$ |

### 3.3. Calculation Method of TTTRR-Type five-Axis Machine Tools

This section introduces the mapping expression for sensitive PDGEs of the TTTRR-type five-axis reconfigurable machine tool. According to this mapping expression, sensitive PDGEs can be quickly calculated without error modeling and analysis.

The rotational axis closest to the tool is marked as $R_1$, the rotational axis farther from the tool is marked as $R_2$, and the axis vector of tool is $z$ direction. The meaning of the symbols in the mapping expression is the same as in Section 3.2; the specific mapping expression is as follows.

### 3.3.1. Sensitive Linear PDGEs of Translational Axis

If the sensitive linear PDGEs of translational axes are identified, $i = 1$, $Axis \in (X, Y, Z)$ can be obtained according to Section 3.1. Where $Dir \in (x, y, z)$, then the mapping expression is shown as Equation (7).

### 3.3.2. Sensitive Linear PDGEs of Rotational Axes

If the sensitive linear PDGEs of rotational axes are identified, $i = 2$, $Axis \in (A, B, C)$ can be obtained according to Section 3.1. Where $Dir \in (x, y, z)$, then the mapping expression is given as:

(1)　When $Axis = R_1$

$$\begin{cases} \text{if} : Cal(Dir) = Cal(R_2), \text{then} : Typ(2)_{Axis}^{Dir} = \Delta Cal\left(\overline{R_1}\right)(R_1) \\ \text{if} : Cal(Dir) \neq Cal(R_2), \text{then} : Typ(2)_{Axis}^{Dir} = \Delta Cal(X + Y + Z)(R_1) \end{cases} \tag{13}$$

(2)　When $Axis = R_2$

$$\begin{cases} \text{if} : Cal(Dir) = Cal(R_2), \text{then} : Typ(2)_{Axis}^{Dir} = \Delta Cal(R_2)(R_2) \\ \text{if} : Cal(Dir) \neq Cal(R_2), \text{then} : Typ(2)_{Axis}^{Dir} = \Delta Cal\left(\overline{R_2}\right)(R_2) \end{cases} \tag{14}$$

### 3.3.3. Sensitive Angular PDGEs of Translational Axis

If the sensitive linear PDGEs of rotational axes are identified, $i = 3$, $Axis \in (X, Y, Z)$ can be obtained according to Section 3.1. Where $Dir \in (x, y, z)$, then the mapping expression is given as:

$$Typ(3)_{Axis}^{Dir} = \Delta Cal\left(\overline{Dir}\right)(Axis) \tag{15}$$

### 3.3.4. Sensitive Angular PDGEs of Rotational Axis

If the sensitive linear PDGEs of rotational axes are identified, $i = 4$, $Axis \in (A, B, C)$ can be obtained according to Section 3.1. Where $Dir \in (x, y, z)$, then the mapping expression is given as:

(1)　When $Axis = R_1$

$$\begin{cases} \text{if} : Cal(Dir) = Cal(R_2), \text{then} : Typ(4)_{Axis}^{Dir} = \Delta Cal(R_1)(R_1) \\ \text{if} : Cal(Dir) \neq Cal(R_2), \text{then} : Typ(4)_{Axis}^{Dir} = \Delta Cal\left(\overline{Toolvector}\right)(R_1) \end{cases} \tag{16}$$

(2)　When $Axis = R_2$

$$\begin{cases} \text{if} : Cal(Dir) = Cal(R_2), \text{then} : Typ(4)_{Axis}^{Dir} = \Delta Cal(R_1)(R_2) \\ \text{if} : Cal(Dir) \neq Cal(R_2), \text{then} : Typ(4)_{Axis}^{Dir} = \Delta Cal(X + Y + Z)(R_2) \end{cases} \tag{17}$$

### 3.4. Summary of Mapping Expressions

Using the proposed method to compare with the existing literature [19,24], the results of the literature analysis are consistent with the sensitive PDGEs identified by this method. Tables 3 and 4 present a summary of the calculation methods for RTTTR and TTTRR-type five-axis machine tool sensitive PDGEs. Based on these method, sensitive PDGES can be quickly calculated, and it provides guidance for accurate design and error traceability of machine tools.

**Table 3.** Mapping expressions of RTTTR-type five-axis machine tool.

| | Sensitive Linear PDGEs | Sensitive Angular PDGEs |
|---|---|---|
| Translational axis in the workpiece chain | $Typ(1)_{Axis}^{Dir} = \Delta Cal(Dir)(Axis)$ | $Typ(3)_{Axis}^{Dir} = \Delta Cal(\overline{Dir})(Axis)$ |
| Translational axis in the tool chain | $Typ(1)_{Axis}^{Dir} = \Delta Cal(Dir)(Axis)$ | $\begin{cases} \text{if}: Cal(Dir) = Cal(R_2), \\ \text{then}: Typ(3)_{Axis}^{Dir} = \Delta Cal(\overline{R_2})(Axis) \\ \text{if}: Cal(Dir) \neq Cal(R_2), \\ \text{then}: Typ(3)_{Axis}^{Dir} = \Delta Cal(R_2)(Axis) \\ \text{if}: \text{ tool chain has 3 translational axes,} \\ \text{and } Cal(T_F) \neq Cal(R_1 + R_2), \\ \text{then}: Typ(3)_{Axis}^{Dir} = \Delta Cal(\overline{Dir})(T_F) \end{cases}$ |
| Rotational axis in the workpiece chain | $\begin{cases} \text{if}: Cal(Dir) = Cal(Axis), \\ \text{then}: Typ(2)_{Axis}^{Dir} = \Delta Cal(Axis)(Axis) \\ \text{if}: Cal(Dir) \neq Cal(Axis), \\ \text{then}: Typ(2)_{Axis}^{Dir} = \Delta Cal(\overline{Axis})(Axis) \end{cases}$ | $\begin{cases} \text{if}: Cal(Dir) = Cal(R_1), \\ \text{then}: Typ(4)_{Axis}^{Dir} = \Delta Cal(\overline{R_1})(R_1) \\ \text{if}: Cal(Dir) \neq Cal(R_1), \\ \text{then}: Typ(4)_{Axis}^{Dir} = \Delta Cal(X+Y+Z)(R_1) \end{cases}$ |
| Rotational axis in the tool chain | $\begin{cases} \text{if}: Cal(Dir) = Cal(Axis), \\ \text{then}: Typ(2)_{Axis}^{Dir} = \Delta Cal(Axis)(Axis) \\ \text{if}: Cal(Dir) \neq Cal(Axis), \\ \text{then}: Typ(2)_{Axis}^{Dir} = \Delta Cal(\overline{Axis})(Axis) \end{cases}$ | $\begin{cases} \text{if}: Cal(Dir) = Cal(Toolvector), \\ \text{then}: Typ(4)_{Axis}^{Dir} = \Delta Cal(R_2)(R_2) \\ \text{if}: Cal(Dir) \neq Cal(Toolvector), \\ \text{then}: Typ(4)_{Axis}^{Dir} = \Delta Cal(\overline{Dir+Toolvector})(R_2) \end{cases}$ |

**Table 4.** Mapping expressions of TTTRR-type five-axis machine tool.

| | Sensitive Linear PDGEs | Sensitive Angular PDGEs |
|---|---|---|
| Translational axis in the workpiece chain | $Typ(1)_{Axis}^{Dir} = \Delta Cal(Dir)(Axis)$ | $Typ(3)_{Axis}^{Dir} = \Delta Cal(\overline{Dir})(Axis)$ |
| Translational axis in the tool chain | $Typ(1)_{Axis}^{Dir} = \Delta Cal(Dir)(Axis)$ | $Typ(3)_{Axis}^{Dir} = \Delta Cal(\overline{Dir})(Axis)$ |
| Rotational axis in the tool chain | $\text{When } Axis = R_1:$ $\begin{cases} \text{if}: Cal(Dir) = Cal(R_2), \\ \text{then}: Typ(2)_{Axis}^{Dir} = \Delta Cal(\overline{R_1})(R_1) \\ \text{if}: Cal(Dir) \neq Cal(R_2),' \\ \text{then}: Typ(2)_{Axis}^{Dir} = \Delta Cal(X+Y+Z)(R_1) \end{cases}$ $\text{When } Axis = R_2:$ $\begin{cases} \text{if}: Cal(Dir) = Cal(R_2), \\ \text{then}: Typ(2)_{Axis}^{Dir} = \Delta Cal(R_2)(R_2) \\ \text{if}: Cal(Dir) \neq Cal(R_2), \\ \text{then}: Typ(2)_{Axis}^{Dir} = \Delta Cal(\overline{R_2})(R_2) \end{cases}$ | $\text{When } Axis = R_1:$ $\begin{cases} \text{if}: Cal(Dir) = Cal(R_2), \\ \text{then}: Typ(4)_{Axis}^{Dir} = \Delta Cal(R_1)(R_1) \\ \text{if}: Cal(Dir) \neq Cal(R_2), \\ \text{then}: Typ(4)_{Axis}^{Dir} = \Delta Cal(\overline{Toolvector})(R_1) \end{cases}$ $\text{When } Axis = R_2:$ $\begin{cases} \text{if}: Cal(Dir) = Cal(R_2), \\ \text{then}: Typ(4)_{Axis}^{Dir} = \Delta Cal(R_1)(R_2) \\ \text{if}: Cal(Dir) \neq Cal(R_2), \\ \text{then}: Typ(4)_{Axis}^{Dir} = \Delta Cal(X+Y+Z)(R_2) \end{cases}$ |

## 4. Simulation Analysis

The impeller is a typical part in five-axis machining. In order to verify the accuracy and effectiveness of this calculation method, an impeller was simulated and analyzed. The value of PDGEs were preset, the linear PDGEs were set as 0.01 mm × sin(0.5 × position), and the angular PDGEs were set as 0.01 deg × sin(0.5× angle). The value range of PDGEs was borrowed from literature [12,14,15,28].

VERICUT software was used to simulate the machining of the impeller, and the machining errors before and after compensation were compared. Firstly, 100 analysis points were selected on the surface of one side of the impeller model, as shown in Figure 4a. Then, based on the sensitive geometric error analysis results in Table 2, the actual inverse kinematics [28] were used to calculate the NC codes after compensation in each direction.

Taking the analysis results in *x* direction as an example, Figure 4b shows the impeller processing diagram after compensating for the sensitive PDGEs. Figure 5 shows some NC codes before and after compensating for the sensitive PDGEs in *x* direction. Figure 6a shows the theoretical error and the machining error before and after compensating for the sensitive PDGEs in *x* direction; the values in Figure 6 are the maximum machining error before and after compensation. The machining error before and after compensation in the *y* and *z* direction sensitive PDGEs are shown in Figure 6b,c, respectively.

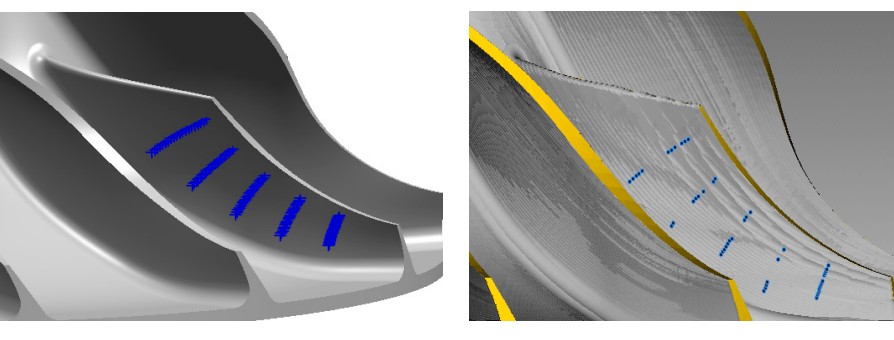

(**a**)Theoretical model of impeller      (**b**)Simulation model of impeller

**Figure 4.** Impeller model diagram.

N8369 G90 G01 X-10.1404 Y-196.0379 Z-132.9920 A-77.5201 C356.5274 F500;
N8370 G90 G01 X-10.2609 Y-195.8336 Z-132.0585 A-77.1917 C356.3618 F500;
N8371 G90 G01 X-10.2142 Y-195.7110 Z-130.9922 A-76.7751 C356.1728 F500;
N8372 G90 G01 X-10.5037 Y-195.2733 Z-129.4145 A-76.2347 C355.8905 F500;
N8373 G90 G01 X-10.7812 Y-194.8322 Z-127.8905 A-75.7112 C355.6166 F500;
N8374 G90 G01 X-10.9103 Y-194.4712 Z-126.3991 A-75.1691 C355.3483 F500;
N8375 G90 G01 X-11.0496 Y-194.0891 Z-124.9332 A-74.6371 C355.0830 F500;
N8376 G90 G01 X-11.0825 Y-193.7838 Z-123.6034 A-74.1338 C354.8414 F500;
N8377 G90 G01 X-11.2087 Y-193.3548 Z-122.0680 A-73.5677 C354.5596 F500;
N8378 G90 G01 X-11.3481 Y-192.9289 Z-120.6562 A-73.0494 C354.2988 F500;
N8379 G90 G01 X-11.4944 Y-192.4607 Z-119.1706 A-72.5006 C354.0217 F500;

N8369 G90 G01 X-10.1072 Y-196.1347 Z-133.1119 A-77.5406 C356.5495 F500;
N8370 G90 G01 X-10.2249 Y-195.9175 Z-132.1616 A-77.2094 C356.3808 F500;
N8371 G90 G01 X-10.1773 Y-195.6125 Z-130.8724 A-76.7545 C356.1506 F500;
N8372 G90 G01 X-10.4736 Y-195.2522 Z-129.3892 A-76.2304 C355.8858 F500;
N8373 G90 G01 X-10.7429 Y-194.8945 Z-127.9645 A-75.7241 C355.6305 F500;
N8374 G90 G01 X-10.8722 Y-194.4959 Z-126.4281 A-75.1741 C355.3537 F500;
N8375 G90 G01 X-11.0169 Y-194.0969 Z-124.9423 A-74.6387 C355.0847 F500;
N8376 G90 G01 X-11.0521 Y-193.6983 Z-123.5054 A-74.1166 C354.8227 F500;
N8377 G90 G01 X-11.1760 Y-193.2526 Z-121.9524 A-73.5473 C354.5373 F500;
N8378 G90 G01 X-11.3194 Y-192.8376 Z-120.5541 A-73.0314 C354.2790 F500;
N8379 G90 G01 X-11.4605 Y-192.3892 Z-119.0917 A-72.4865 C354.0064 F500;

(**a**)Before compensation          (**b**)After compensation

**Figure 5.** Part of the NC codes.

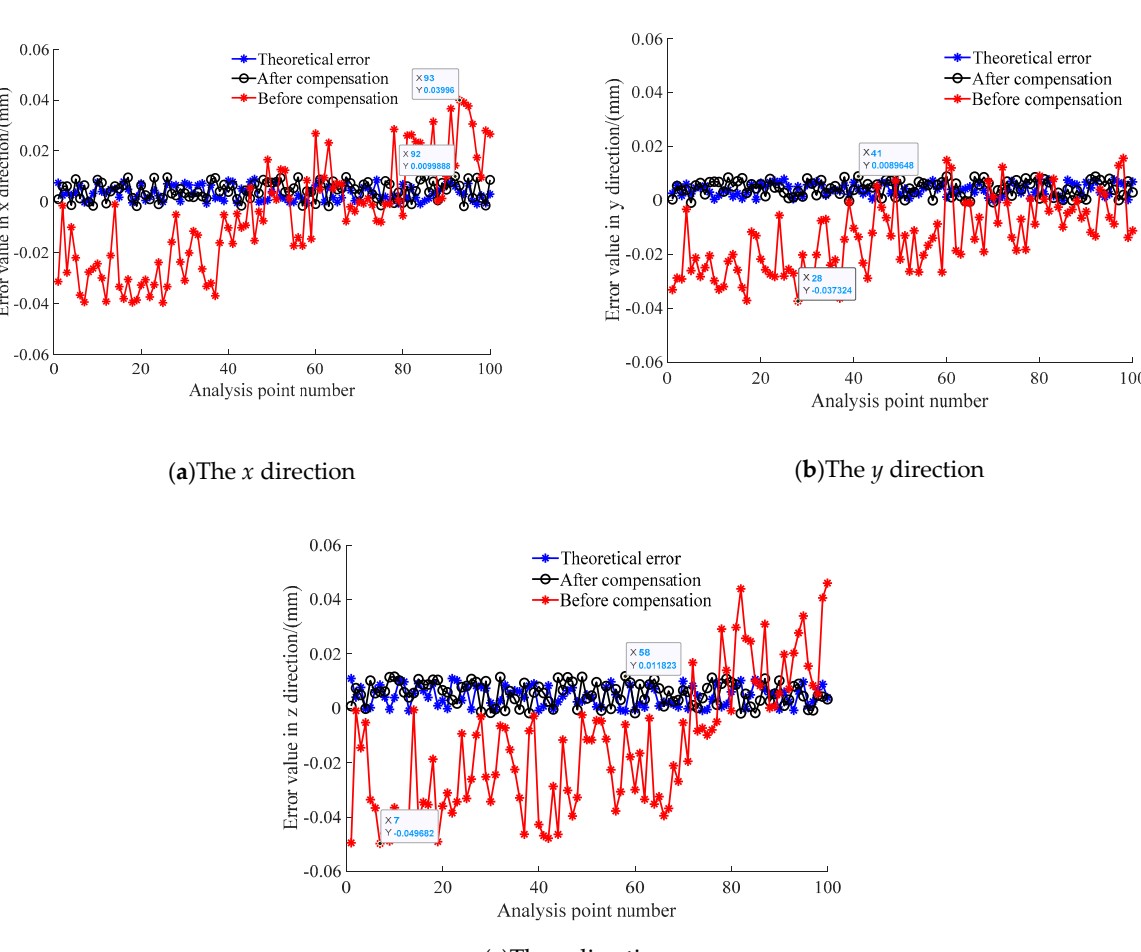

(**a**)The *x* direction          (**b**)The *y* direction

(**c**)The *z* direction

**Figure 6.** Comparison of results before and after compensation in each direction.

It can be found that the machining error after compensating for these sensitive PDGEs is almost consistent with the theoretical error, which indicates that the machining accuracy is mainly affected by these sensitive PDGEs. At the same time, the machining error after compensation is significantly lower than that before processing, which indicates the accuracy of the proposed method.

According to the proposed method, Matlab software was used to make related software. As shown in Figure 7, the sensitivity index of each PDGE was calculated through the function embedded in the software by inputting the topology of the machine tool, and the sensitive PDGEs were displayed.

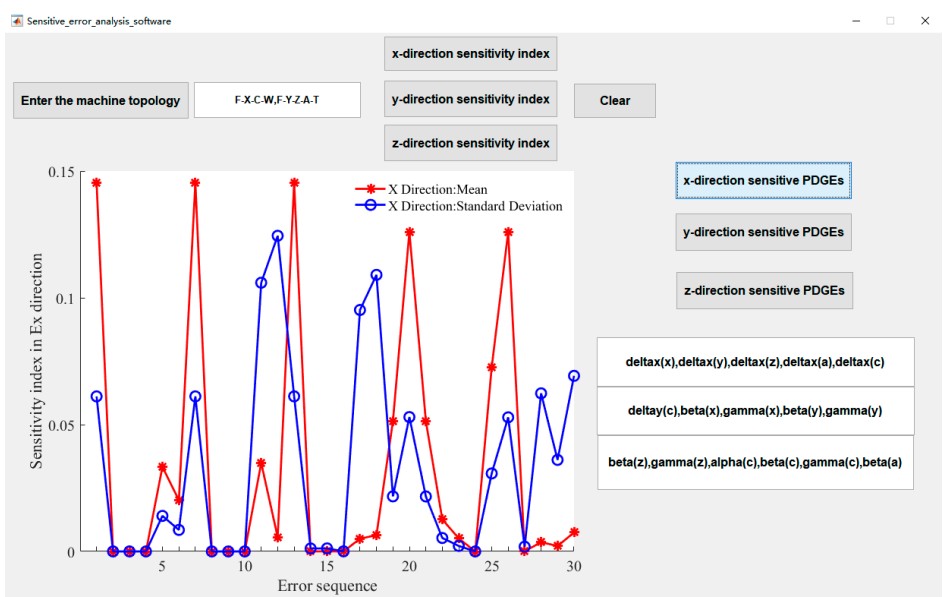

**Figure 7.** Sensitive PDGEs calculation software.

## 5. Conclusions

A new high-efficient calculation method for sensitive PDGE identification in the five-axis reconfigurable machine tool was proposed and verified in this paper. According to machine tool configuration axis information, sensitive PDGEs of RTTTR and TTTRR-type five-axis reconfigurable machine tools can be quickly identified according to the proposed method without traditional HTM modeling and sensitivity analysis.

The simulation results show that the machining accuracy of the impeller is significantly improved after compensating for sensitive PDGEs. According to the simulation results, the proposed method is accurate. It can provide a reference for precision design and error compensation of five-axis reconfigurable machine tools.

Based on the proposed calculation method, the barrier-free output of sensitive PDGEs can be realized through the programming of related software. It can quickly respond to the sensitive PDGEs identification in the reconstructed five-axis machine tool. Rapid identification of sensitive PDGEs for five-axis reconfigurable machine tools is studied in this paper. In future work, the calculation methods of other error sources of five-axis reconfigurable machine tools are also worth further study.

**Author Contributions:** Z.S. contributed to the conception of the study. S.D. helped perform the analysis with constructive discussions. Z.C., and Z.L. performed the simulation. Z.S., and Z.W. contributed significantly to manuscript preparation. All authors have read and approved the manuscript being submitted and agree to its submission to this journal.

**Funding:** National Natural Science Foundation of China (No. 51905470), China Postdoctoral Science Foundation (No. 2020M671617), Natural Science Foundation of the Jiangsu Higher Education

Institutions of China (No. 19KJB460029), and Jiangsu Postgraduate Research Innovation Project of China (No. SJCX20_1373).

**Data Availability Statement:** The datasets used or analyzed during the current study are available from the corresponding author on reasonable request.

**Acknowledgments:** Thanks for the support of Mechanical Engineering College of Yangzhou University.

**Conflicts of Interest:** The authors declare no conflict of interest.

## Appendix A

| Linear PDGEs | | Angular PDGEs | |
|---|---|---|---|
| $\Delta Cal(parm)(X)$ | $\Delta Cal(\overline{parm})(X)$ | $\Delta Cal(parm)(X)$ | $\Delta Cal(\overline{parm})(X)$ |
| $\Delta Cal(x)(X)= \Delta x(X)$ | $\Delta Cal(\overline{x})(X)= \{\Delta y(X),\Delta z(X)\}$ | $\Delta Cal(x)(X)= \Delta\alpha(X)$ | $\Delta Cal(\overline{x})(X)= \{\Delta\beta(X),\Delta\gamma(X)\}$ |
| $\Delta Cal(y)(X)= \Delta y(X)$ | $\Delta Cal(\overline{y})(X)= \{\Delta x(X),\Delta z(X)\}$ | $\Delta Cal(y)(X)= \Delta\beta(X)$ | $\Delta Cal(\overline{y})(X)= \{\Delta\alpha(X),\Delta\gamma(X)\}$ |
| $\Delta Cal(z)(X)= \Delta z(X)$ | $\Delta Cal(\overline{z})(X)= \{\Delta x(X),\Delta y(X)\}$ | $\Delta Cal(z)(X)= \Delta\gamma(X)$ | $\Delta Cal(\overline{z})(X)= \{\Delta\alpha(X),\Delta\beta(X)\}$ |
| $\Delta Cal(X)(X)= \Delta x(X)$ | $\Delta Cal(\overline{X})(X)= \{\Delta y(X),\Delta z(X)\}$ | $\Delta Cal(X)(X)= \Delta\alpha(X)$ | $\Delta Cal(\overline{X})(X)= \{\Delta\beta(X),\Delta\gamma(X)\}$ |
| $\Delta Cal(Y)(X)= \Delta y(X)$ | $\Delta Cal(\overline{Y})(X)= \{\Delta x(X),\Delta z(X)\}$ | $\Delta Cal(Y)(X)= \Delta\beta(X)$ | $\Delta Cal(\overline{Y})(X)= \{\Delta\alpha(X),\Delta\gamma(X)\}$ |
| $\Delta Cal(Z)(X)= \Delta z(X)$ | $\Delta Cal(\overline{Z})(X)= \{\Delta x(X),\Delta y(X)\}$ | $\Delta Cal(Z)(X)= \Delta\gamma(X)$ | $\Delta Cal(\overline{Z})(X)= \{\Delta\alpha(X),\Delta\beta(X)\}$ |
| $\Delta Cal(x+y)(X)= \{\Delta x(X),\Delta y(X)\}$ | $\Delta Cal(\overline{x+y})(X)= \Delta z(X)$ | $\Delta Cal(x+y)(X)= \{\Delta\alpha(X),\Delta\beta(X)\}$ | $\Delta Cal(\overline{x+y})(X)= \Delta\gamma(X)$ |
| $\Delta Cal(x+z)(X)= \{\Delta x(X),\Delta z(X)\}$ | $\Delta Cal(\overline{x+z})(X)= \Delta y(X)$ | $\Delta Cal(x+z)(X)= \{\Delta\alpha(X),\Delta\gamma(X)\}$ | $\Delta Cal(\overline{x+z})(X)= \Delta\beta(X)$ |
| $\Delta Cal(y+z)(X)= \{\Delta y(X),\Delta z(X)\}$ | $\Delta Cal(\overline{y+z})(X)= \Delta x(X)$ | $\Delta Cal(y+z)(X)= \{\Delta\beta(X),\Delta\gamma(X)\}$ | $\Delta Cal(\overline{y+z})(X)= \Delta\alpha(X)$ |
| $\Delta Cal(x+y+z)(X)= \{\Delta x(X),\Delta y(X),\Delta z(X)\}$ | | $\Delta Cal(x+y+z)(X)= \{\Delta\alpha(X),\Delta\beta(X),\Delta\gamma(X)\}$ | |
| $\Delta Cal(X+Y)(X)= \{\Delta x(X),\Delta y(X)\}$ | $\Delta Cal(\overline{X+Y})(X)= \Delta z(X)$ | $\Delta Cal(X+Y)(X)= \{\Delta\alpha(X),\Delta\beta(X)\}$ | $\Delta Cal(\overline{X+Y})(X)= \Delta\gamma(X)$ |
| $\Delta Cal(X+Z)(X)= \{\Delta x(X),\Delta z(X)\}$ | $\Delta Cal(\overline{X+Z})(X)= \Delta y(X)$ | $\Delta Cal(X+Z)(X)= \{\Delta\alpha(X),\Delta\gamma(X)\}$ | $\Delta Cal(\overline{X+Z})(X)= \Delta\beta(X)$ |
| $\Delta Cal(Y+Z)(X)= \{\Delta y(X),\Delta z(X)\}$ | $\Delta Cal(\overline{Y+Z})(X)= \Delta x(X)$ | $\Delta Cal(Y+Z)(X)= \{\Delta\beta(X),\Delta\gamma(X)\}$ | $\Delta Cal(\overline{Y+Z})(X)= \Delta\alpha(X)$ |
| $\Delta Cal(X+Y+Z)(X)= \{\Delta x(X),\Delta y(X),\Delta z(X)\}$ | $\Delta Cal(\overline{A})(X)= \{\Delta y(X)\,\Delta z(X)\}$ | $\Delta Cal(X+Y+Z)(X)= \{\Delta\alpha(X),\Delta\beta(X),\Delta\gamma(X)\}$ | $\Delta Cal(\overline{A})(X)= \{\Delta\beta(X),\Delta\gamma(X)\}$ |
| $\Delta Cal(A)(X)= \Delta x(X)$ | $\Delta Cal(\overline{B})(X)= \{\Delta x(X),\Delta z(X)\}$ | $\Delta Cal(A)(X)= \Delta\alpha(X)$ | $\Delta Cal(\overline{B})(X)= \{\Delta\alpha(X),\Delta\gamma(X)\}$ |
| $\Delta Cal(B)(X)= \Delta y(X)$ | $\Delta Cal(\overline{C})(X)= \{\Delta x(X),\Delta y(X)\}$ | $\Delta Cal(B)(X)= \Delta\beta(X)$ | $\Delta Cal(\overline{C})(X)= \{\Delta\alpha(X),\Delta\beta(X)\}$ |
| $\Delta Cal(C)(X)= \Delta z(X)$ | | $\Delta Cal(C)(X)= \Delta\gamma(X)$ | |
| $\Delta Cal(A+B)(X)= \{\Delta x(X),\Delta y(X)\}$ | | $\Delta Cal(A+B)(X)= \{\Delta\alpha(X),\Delta\beta(X)\}$ | |
| $\Delta Cal(A+C)(X)= \{\Delta x(X),\Delta z(X)\}$ | | $\Delta Cal(A+C)(X)= \{\Delta\alpha(X),\Delta\gamma(X)\}$ | |
| $\Delta Cal(B+C)(X)= \{\Delta y(X),\Delta z(X)\}$ | | $\Delta Cal(B+C)(X)= \{\Delta\beta(X),\Delta\gamma(X)\}$ | |

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
