# Peer review of "High-Efficient Calculation Method for Sensitive PDGEs of Five-Axis Reconfigurable Machine Tool"

_machines, doi:10.3390/machines9050084_

Round 1
Reviewer 1 Report
The article proposes a highly efficient calculation method for position-dependent sensitive geometrical errors (PDGE) of re-configurable five-axis machine tools.
The study of geometric errors on 5-axis machines is a current and very important topic in the manufacture of metal parts.
However, the article is confusing and difficult to read. The contribution or what is new in the article is not clearly defined, intuition must be used and this is not appropriate for a scientific article. The introduction is a description of many methods to analyse sensitive geometrical errors of machine tools, it is very generic.
The introduction is not related to the specific objective of the article, what is the contribution of the article to the state of the art?
In section 2 the Morris global sensitivity analysis method is described, this is widely known, what is new and important?
Sections 3 High-efficient calculation method for sensitive PDGEs of five-axis re-configurable machine tool and 4 Application of calculation method, require major changes. An effort should be made to explain these sections in a different way.
The Simulation Analysis section shows the result of a PDGE applied to a machine. What is the difference between this method and the Morris method?
The Conclusions section shows results rather than conclusions.
The article needs major changes in order to be accepted.
Reviewer 2 Report
The paper presents a method to evaluate and compensate position dependant geometric error on five axis machines. The proposed methods aims to improve the computation for reconfigurable machine tools. The subject is interesting but the paper fails to provide sufficient information to be check or reused by other researchers. Main concerns are listed below:
for five axis machines (where rotary axis are involved), there can be nonlinearties involved in the link between input (angles) and output (position), so the sensitivities computed using equation (4) can be position-dependant. For what particular configuration has the problem been solved and do the authors have a comment on the generality of the approach ?
The criteria that has been used (equation 3) only deals with end-effector position, however in 5 axis machining, effector orientation is also an important criteria to be considered, do the authors have a comment on this ?
algorithm showed in figure 3(b) needs more explanation, what is the meaning of 'no modelling required' under step 1? Isn't HTM necessary to determine at this stage ?
paper give a lot of definition on part 3.2.1 but not the actual way to compute all those quatities, an explanation or a reference is at least needed here
on part 5, it is stated that a compensation is used, but no details is given on that point. Practically speaking, how is it possible to create thus compensation and is it applicable on real case? Is it an action on the numerical control or on the machine tool structure itself?
the paper presents some results for the sensitivities for a particular example, however, the data are not provided, so it is difficult for the reviewer to cross check the results. What is that machine that has been modelled for those results (figure 2 for example) ?
the paragraph on lines 168--173 is not clear. Are these experimental on numerical results and why chosing 120 cycles (and not more or less) for statistical analysis ? a convergence graph should be provided for example
on figure 2, what is the explanation of the null values of sesititities for some of the case that have been treated ? Is there a physical meaning behind that ?
In addition, some details may be check during revision of the paper:
it is stated in the paper that table 1 shows 30 PDGEs, however, the table mention 36 terms, so it is a bit confusing
formally speaking, second part of equation (5) give a biased estimator of the standard deviation because µ is an estimator of the mean an not the actual mean, the division must be performed by SN-1 instead of SN
there are some formatting error eading to misalignment of formulas on line 234-245
Reviewer 3 Report
The authors raised a significant problem in machining. A very important and important topic. The article is written in everyday language. Inability to understand the problem at hand. The authors use markings that are understandable only to them. I believe the value of the article will increase as the authors correct the description again. Please describe the algorithm of operation when evaluating machine tool errors. Too many markings in the text - nothing comes of it. I will not mention the conclusions. I believe the article is to be rejected as it stands.
Round 2
Reviewer 1 Report
The changes introduced are adequate. The article is much improved.
Reviewer 2 Report
The reviewer thank the authors for their response to comments.
Reviewer 3 Report
Dear Authors.
The suggested modifications have been satisfactorily made.
King regards.